# Next Generation Immuno-Oncology Strategies: Unleashing NK Cells Activity

**DOI:** 10.3390/cells11193147

**Published:** 2022-10-06

**Authors:** Alberto Mendoza-Valderrey, Maite Alvarez, Andrea De Maria, Kim Margolin, Ignacio Melero, Maria Libera Ascierto

**Affiliations:** 1Rosalie and Harold Rae Brown Cancer Immunotherapy Research Program, Borstein Family Melanoma Program, Translational Immunology Department, Saint John’s Cancer Institute, Santa Monica, CA 90404, USA; 2Program for Immunology and Immunotherapy, CIMA, Universidad de Navarra, 31008 Pamplona, Spain; 3Navarra Institute for Health Research (IdiSNA), 31008 Pamplona, Spain; 4Centro de Investigación Biomédica en Red de Cáncer (CIBERONC), 28029 Madrid, Spain; 5Department of Health Sciences, University of Genoa, 16126 Genova, Italy; 6IRCCS Ospedale Policlinico San Martino, 16132 Genova, Italy; 7Borstein Family Melanoma Program, Saint John’s Cancer Institute, Santa Monica, CA 90404, USA; 8Department of Immunology and Immunotherapy, Clínica Universidad de Navarra, 31008 Pamplona, Spain

**Keywords:** natural killer cells, innate immunity, cancer immunotherapy, antibody-based therapeutics, cytokine therapy, NK cell-based therapy

## Abstract

In recent years, immunotherapy has become a powerful therapeutic option against multiple malignancies. The unique capacity of natural killer (NK) cells to attack cancer cells without antigen specificity makes them an optimal immunotherapeutic tool for targeting tumors. Several approaches are currently being pursued to maximize the anti-tumor properties of NK cells in the clinic, including the development of NK cell expansion protocols for adoptive transfer, the establishment of a favorable microenvironment for NK cell activity, the redirection of NK cell activity against tumor cells, and the blockage of inhibitory mechanisms that constrain NK cell function. We here summarize the recent strategies in NK cell-based immunotherapies and discuss the requirement to further optimize these approaches for enhancement of the clinical outcome of NK cell-based immunotherapy targeting tumors.

## 1. Introduction to NK Cell Biology

### 1.1. NK Cells, MHC Class I and the Missing Self

A basic truth about scientific research is that we cannot always predict where it is going to lead. No one knows this better than immunologist Rolf Kissling, who, by intending to study T cell responses to cancer, discovered a small subset of lymphocytes in 1975, termed natural killer (NK) cells, which are able to directly kill tumor cells without need for pre-immunization or antigen specificity [1]. NK cells represent 5–15% of circulating lymphocytes in humans and have historically been described as CD3^−^CD14^−^CD56^+^ cells with large granuli. However, NK cells can be also found in the liver, spleen, bone marrow, and lymph nodes, and they are characterized by heterogenous receptor expression patterns and functional specificities based on their tissue localization. Despite this heterogeneity, all NK cells are characterized by many inhibitory and activating receptors, whose dynamic balance determines their activity against the target cells [2] (Figure 1). 

Inhibitory receptors, such as killer immunoglobulin-like receptors (KIRs) and the CD94-NKG2A heterodimer receptors, deliver negative signals that prevent NK cell autoreactivity. Those receptors recognize molecules of the major histocompatibility complex (MHC) class I, whose absence will result in NK activation. Several studies have shown that for a proper activation of NK cells in case of lack of MHC expression, signaling from NK activating receptors, such as NKG2D, NKG2C, NKp46, NKp30, NKp44, and CD226, is also necessary. In physiologic conditions, a ‘self-recognition’ phenomenon is established, resulting from the fact that the binding of NK inhibitory receptors with self MHC class I allows the downregulation of the activating receptors and the tolerance of NK cells to self-tissue. In non-physiologic conditions, such as the development of tumor malignancies or bacterial/viral infections, a ‘missing self’ phenomenon occurs instead. This phenomenon is usually characterized by a downregulation of MHC class I molecules on the target, a blockage of inhibitory signals and the induction of NK cell activating counterreceptors, and the unleashing of cytotoxic effects. Two main mechanisms lead to NK- mediated cytotoxic activity of target cells: (i) the release of cytoplasmic granules which contains perforin and granzyme; and (ii) the expression of tumor necrosis factor (TNF) family members, including FASL or TNF-related apoptosis-inducing ligand (TRAIL), which promote target cell apoptosis by interacting with their corresponding counterreceptors. In addition, most NK cells express the low-affinity activating receptor FcγRIIIa (CD16) that binds the Fc portion of immunoglobulin G1 (IgG1) and mediates antibody-dependent cellular cytotoxicity (ADCC), resulting in NK cell degranulation against antibody-coated target cells (Figure 2). NK cells with high expression of CD16 are known as CD56 dim, and are characterized by the highest cytotoxicity and ADCC activities. While predominant in the blood, only a small number of CD56 dim NK is observed in lymph nodes and other tissues. On the contrary, tissue-resident NK cells are mainly enriched in a subset of NK cells, known as CD56 bright, which are capable of releasing pro-inflammatory cytokines and chemokines that could promote direct anti-tumor activity, beyond innate and adaptive responses [3,4]. This evidence indicates that NK cells, apart from being killers, are also immunoregulatory cells that can modulate immune responses through cross-talking with additional immune cells. 

### 1.2. NK Cell Immune Crosstalk

NK cells interact with several other cells of the immune system, most notably DCs, myeloid cells, and T cells. The complex and bidirectional interactions occurring between NK cells and dendritic cells (DCs) are the subject of intense research. Historically, the main role of DCs was characterized by their capacity to prime antigen-specific immune responses, but their contribution to innate immunity is currently accepted. Various contexts have documented interactions between DCs and NK cells, with emerging evidence of complex bidirectional crosstalk between both cell types [5,6]. Much evidence supports the idea that these interactions not only set the stage for immune responses, but also determine the functional outcomes. DCs can impact NK cell functions by inducing their activation and/or proliferation. DC-mediated activation of NK cells, which leads to raised NK cell cytolytic activity and/or interferon-γ (IFN-γ) production, can be triggered by both resting and activated DCs. Both cell-contact-dependent interactions and soluble cytokines are involved in this process. Particularly, the NK group 2, member (NKG2D) ligands expressed by DCs, adhesion molecules including LFA1 (lymphocyte function-associated antigen 1), and the cytokines derived from DCs have been involved in DC-mediated proliferation of NK cells. Conversely, NK cells can influence DC functions and induce DC maturation, activation, or elimination. While the NK cell mediated elimination of DC is mainly dependent on NKp30 receptors recognizing B7-H6, NK cell-mediated DC activation appears to be mostly dependent on cytokines. In particular, NK cells can recruit conventional type 1 dendritic cells (cDC1) through the release of chemokines and cytokines, including CCL5 (RANTES), XCL-1/XCL-2, and FLT3 ligand [5]. Several reports have demonstrated that the expression of the chemokine XCL1 by NK cells plays an important role in the tumor microenvironment (TME) recruitment of XCR1^+^ type 1 DCs (cDC1), which is a cell subtype crucial for the orchestration of an effective CD8^+^ T cell-mediated tumor immune response, due to the excellence of its performance at antigen cross-presentation [6,7]. These DCs can then migrate to lymph nodes, mediating T cell activation and expansion, as well as the recruitment of effector lymphocytes to the site of infection or tumor. 

NK cells can also influence DC maturation by means of secretion of granulocyte-macrophage colony-stimulating factors (GM-CSF), interferon-γ, and TNFα [8,9,10]. TNF-TNFR2 signaling seems to have a prominent role in the modulation of interactions between both DC and NK cells. Tufa and collaborators showed that after TLR4 stimulation, a particular myeloid DC subset, the 6-sulfo LacNAc DC (slanDC), increased the expression of transmembrane TNF that leads to NK cell activation with upregulation of TNFR2 [11]. Interestingly, GM-CSF further supports DC activation and secretion of IL-12, which increases NK cell production of IFNγ and cytotoxicity [11]. The importance of TNF-TNFR2 signaling in DC-NK cell crosstalk to achieve a powerful anti-tumor response has been highlighted in several studies targeting the TNFR2 signaling pathway [5,12,13]. Additional members of the TNFR family, such as 4-1BB (CD137), might also be important in mediating anti-tumor response by acting on NK:DC crosstalk [14,15]. These concepts, however, should be better explored from a clinical point of view. Notably, environmental factors, such as transforming growth factor (TGF)-β and prostaglandin E2 (PGE_2_) production, induce NK cell-cDC1 axis avoidance by impairing NK cell viability and downregulating chemokine receptor expression in cDC1 [16]. Cross-regulation between NK cells and myeloid cells, particularly macrophages, has also been a recent topic of interest. Over the course of viral infection or cancer initiation, M0 and M2 macrophages become polarized towards the M1 phenotype soon after phagocyting cancer cells or infected cells. M1 macrophages release proinflammatory cytokines such as IL-12, IL-18, TNF, type I IFN, and CCR7, which activate NK cells. When activated, NK cells eliminate M0 and M2 macrophages, which are characterized by low expression of HLA class I, while M1 remains resistant to NK attacks by keeping upregulated MHC class I on their surface. Apart from the regulation of myeloid cells by NK cells, data regarding the collaboration of NK cells with T cells are emerging, indicating a bidirectional crosstalk between both cell types. Through the secretion of IL-2 and IL-15, T cells activate NK cells. On the other hand, NK cells can help T cell activation and T proliferation into cytotoxic T lymphocyte (CTL) by producing IFNg and IL-12. Notably, IL-10 and DNAX accessory molecule-1 (DNAM-1, also known as CD226) secreted by NK cells can also inhibit T cell response via deletion of CTL, thus preventing the phenomenon of autoimmunity.

## 2. NK Cells and Cancer Therapy

### 2.1. Impact of NK Cells on Cancer Patient’s Prognosis 

NK cells can conceivably play a fundamental role in cancer patient outcomes. Different studies have evidenced that the presence of altered circulating NK cell subpopulations and the increase in the levels of some NK cell subsets (absolute number and percentage) are directly correlated with the prognoses of cancer patients (Table 1). However, some of these studies are limited by the small number or patients included and the short patient follow-up, whereby the potential NK cell prognostic significance for long-term survival requires further investigation [17,18]. In addition, some studies have reported conflicting results depending on the cancer type. While circulating NKp46^+^ NK cells have been associated with better prognosis in colorectal cancer (CRC) [19], Picard et al. [20] reported the association of NKp46^+^ NK cells with poor patient survival in Non-Small Cell Lung Cancer (NSCLC) patients, suggesting the ability of these cells to suppress anti-tumor Th1 immunity. Hence, more studies are needed to clarify these observations. Studies performed in tumor biopsy specimens have instead been mainly focused on the assessment of intra-tumoral T lymphocytes and not on NK cells, due to two main factors. The first factor is the reduced number of infiltrated NK cells in the TME of several solid cancers, due to their poor infiltration ability. The second reason is related to the inaccurate markers used to identify tumor-infiltrating NK cells. Indeed, NK cells share multiple common surface molecules also expressed by other tumor-infiltrating lymphoid cells, such as CD57. These factors contribute to the fact that CD8^+^ T cells have garnered most of the attention of the immune-cell analyses performed in the TME. In addition to this, the CD56 marker can also be expressed by tumor cells themselves [21], supporting the inaccurate usage of this marker for a correct NK cell identification in the TME. The identification of the NCR1 (NKp46) marker as an almost exclusive NK cell specific surface receptor [22] instead provides a more reliable quantification of these cells in the TME. However, to date, it is reported that only 9% of the studies assessing tumor cell infiltration used antibodies against this marker to identify NK cells [23]. Once this technical limitation was overcome, several studies associated high intra-tumoral NK cell density with positive prognostic value in different cancer types (Table 1). Conversely, low levels of tumor-infiltrating NK cells in other cancer types, such as prostate [24] and gastric [25] cancers, are linked to a poorer patient prognosis. At a more detailed level, it was reported that the region within the tumor which NK cells infiltrate directly impacts patient outcome, with survival rates higher in the cases in which NK cell infiltration occurs into the cancer nests than in those who had peritumoral or stromal infiltration [23]. Patient prognoses in some cancer types has been found also to be associated with the expression of the activating NK receptor Nkp30 isoforms; while expression of the activating isoforms 1/2 is associated with better patient prognosis, the immunosuppressive NKp30 isoform 3 constitutes a negative prognostic marker for gastrointestinal tumors [26] and high-risk neuroblastoma [27]. In line with this, a decrease in NK cell activity has also been significantly associated with an advanced clinical stage in multiple myeloma patients [28]. In breast cancer, we showed—by analyzing primary breast cancer samples obtained from patients who underwent either ~5–9 years relapse-free survival or developed tumor relapse within ~1–6 years—that the expression of molecules implicated in NK cell activating signaling and in interaction between NK cells and target cells is higher in favorable prognosis patients. Additionally, the expression of isoform 1 of the activating NKp30 showed statistically significant differences upon comparison with the relapse group [29]. Years later, through the development of an eight-gene NK cell-specific transcriptional signature which allowed to estimate the abundance of NK cells in the TME, we showed that while the increased expression of these markers is associated with good melanoma, sarcoma, breast, and endometrial cancer patient prognoses, it is paradoxically associated with worse overall survival in uveal melanoma patients [30]. The presence of CD57^+^ NK cells in CRC TME has also been associated with good prognoses [31]. Similar results were obtained in lung cancer patients [32], as well as regarding the relapse-free survival of HCC patients following surgical resection [33].

### 2.2. NK Cells in the Tumor Microenvironment 

Although NK cells are important mediators of cancer immunosurveillance and tumor rejection, their infiltration, activity, and proliferation in the TME is highly compromised. Multiple negative regulators or phenomena, such as high expression of TFG-β or PGE2, chronic IFN-γ production, high concentration of lactate and adenosine, or overexpression of indoleamine 2,3-dioxygenase (IDO), have been found to lead directly or indirectly to dysfunction or poor infiltration of NK cells in the TME [39,40]. Although the detailed review of these mechanisms is beyond the scope of this work, in the next chapter, we will highlight how current immunotherapeutic strategies have been explored to allow proper cytotoxic activity of NK cells against tumor cells. 

### 2.3. Strategies for Restoring NK Cell Anti-Tumor Immunity

In recent years, NK cell-based emerging immunotherapies have undergone significant advances for hematological malignances, particularly acute myeloid leukemia (AML). However, their efficacy against solid tumors remains limited due to the low capacity of NK cells to infiltrate them, the short survival time of NK cells, and the impact exerted by the immunosuppressive milieu of the TME in regulating NK cell cytotoxicity [41]. Below, some of the most important strategies developed to enhance NK cells’ cytotoxic properties and improve their anti-tumor efficacy are described. These can be broadly divided into (1) antibody-based therapeutics, (2) cytokines therapy, and (3) NK cell-based transfer therapy. 

#### 2.3.1. Unleashing NK Cell Activity by Antibody-Based Therapeutics 

Nowadays, the usage of therapeutic antibodies (Abs) represents the main targeted approach for the treatment of cancer. Therapeutic antibodies that can modulate NK cell activity can be classified into three different groups: (a) monoclonal or bispecific Abs (mAbs or bsAbs) directly targeting NK cell receptors (e.g., anti KIRs, anti-PD-(L)1, anti- NKG2A, anti-TIGIT, anti NKG2D and 2B4), (b) killer cell engagers able to target NK and tumor cells receptors (e.g., CD16-CD33 bispecific killer cell engager, BiKE; or CD16-CD19-NKp46 trispecific killer cell engager, TriKE), and (c) Abs which do not act directly on NK cell receptors, but are able to modify the cytotoxicity of NK cells through an effect which, most likely, affects the ADCC capacity (e.g., anti-EGFR, anti-HER-2, etc.).

#### Therapeutic Antibodies Directly Targeting NK Cell Receptors

NK cells express many inhibitory receptors or immune checkpoint receptors, some of which have been targeted by cancer immunotherapy and are described below.

PD-1 and PD-L1: PD1 (programmed cell death protein 1) is an immune checkpoint receptor mainly expressed on T cells. The intersection of the PD-1 receptor with the ligands PD-L1 (also known as B7-H1 and CD274) and PDL2 (also known as B7-DC and CD273) regulates immune cell activation, causing the suppression of immune cell proliferation and the decrease in cytokine production [42]. This interaction is fundamental for restriction of T cell activity in peripheral tissues during inflammatory responses to infection, as well as for prevention of the phenomenon of autoimmunity. Unfortunately, in order to escape the action of the immune system and to continue to proliferate, a multitude of tumors overexpress the ligands of the programmed death 1 (PD-1). This represents a major immune resistance mechanism within the TME, and multiple antibodies have been approved to target the PD-1/PD(L)1 axis in cancer. Currently, the usage of anti PD-(L)1 blocking antibodies has been FDA approved for the treatment of 20 different tumor indications, including melanoma, NSCLC, bladder cancer and triple negative breast cancer (TNBC). PD1 is expressed on a large percentage of CD8+ and CD4+ tumor-infiltrating lymphocytes (TILs). However, several studies are now emerging which show an expression of PD-1 not only on T cells, but also on additional immune cells, including NK cells [43,44] and B cells [45,46,47]. Despite these studies, there is still a lot of controversy about PD-1 expression on NK cells [43,44,48,49,50]. Additionally, PD(L)1 is also expressed on NK cells, likely playing a role in the immunosuppression mediated by the PD-1/PD(L)1 axis [51]. For these reasons, although PD1 inhibition is frequently seen as enhancing the activity of effector T cells in tissues and in the TME, it also likely increases NK cell activity in tumors and tissues (directly or indirectly). In addition, it could also improve antibody production either indirectly or through direct effects on PD1^+^ B cells. It is not surprising to find studies showing an association between NK and B cell infiltration in the TME, or a positive response to immunotherapeutic treatments with anti-PD(L)1 blockade as a backbone [5,30,52,53]. High frequencies of circulating NK cells during anti-PD-1 treatments have been found to be positively correlated with therapeutic benefit in advanced NSCLC patients [54] and melanoma patients [55], respectively. Additionally, in some cases, the higher absolute number of circulating NK cells at treatment baseline has also been correlated with the clinical response [54]. 

High intra-tumoral NK cells are also connected to the response to immunotherapeutic treatments. In pre-treatment tumor biopsies derived from NSCLC patients, the increased expression in the TME of the NK cell-specific transcriptional signature, developed by Ascierto ML et al. [30], together with other genes associated with NK cell priming of the adaptive immune response, correlated with a better response to anti-PD(L)1 immunotherapy. In melanoma patients, the higher levels of intra-tumoral NK cells, which were found to be correlated with intra-tumoral cDC1, increased the responsiveness of patients to anti-PD-1 [5], which suggests that patients with increased intratumoral NK cells have a higher probability of benefiting from anti-PD-1 therapy. In vitro studies have shown that blocking PD1/PD(L)1 signaling enhances activation and cytotoxicity while suppressing apoptosis of NK cells [43]. Additional studies have also confirmed that PD-1/PD(L)1 immunotherapy can elicit NK cell response in several murine cancer models [44], and that the binding of anti-PD(L)1 mAb to PD(L)1^+^ NK cells was indeed responsible for the modulation of their function, exerting an enhanced cytotoxicity against target cells in a process independent of the PD-1/PD(L)1 axis [56]. These results might provide a potential explanation as to why tumors lacking expression of PD(L)1 can respond positively to anti-PD(L)1 mAb therapy [56]. Taken together, this evidence highlights the PD-1/PD-L1 axis to be a fundamental checkpoint for NK cell anti-cancer activity.

TIGIT/CD155/CD112: The immune checkpoint TIGIT (T cell immunoreceptor with immunoglobulin and immunoreceptor tyrosine-based inhibitory motif domains) is expressed on NK cells and other immune types, such as Natural Killer T cells (NKT), known as a mixed group of T cells that exhibit common properties of both T cells and NK cells [57], as well as activated CD4^+^, CD8^+^ T cells [58], and Tregs [59]. Its expression is enhanced during exhaustion [60]. The engagement of the TIGIT receptor with its ligands, CD155 (also known as PVR) and CD112 (also known as PVRL2), suppresses IFNG secretion and inhibits NK cell cytotoxicity [57,61]. In the tumor microenvironment and peripheral blood of cancer patients, TIGIT has usually been found to be co-expressed with PD-1 on CD8^+^ T cells [62]. For this reason, the blockage of PD-1 and TIGIT inhibition is considered an encouraging combination strategy. In animal studies, while each individual inhibition does not significantly hamper the growth of CT26 tumors, the inhibition of TIGIT and PD-1/PD-L1 enhances the activity of anti-tumor CD8^+^ T cells and leads to a complete reduction in tumor size [63]. Remarkably, TIGIT inhibition, both alone and in combination with PD-1 inhibition, mainly acts on NK cells to increase the anti-tumor activity of CD8^+^ T cells. Additionally, NK cell depletion and NK cell-specific TIGIT deficiency jeopardized TIGIT blockade effects [64]. In melanoma patients, results show that the combined PD-1/TIGIT inhibition enhanced in the periphery and the tumor microenvironment the activity of CD8^+^ T cells in comparison to single inhibition [65]. However, whether and how NK cells might lead to a direct enhancement of CD8^+^ T cell activity needs to be carefully assessed. Additionally, in PD-L1-positive NSCLC cancer patients, the combined PD-L1/TIGIT inhibition (atezolizumab + tiragolumab) leads to greater clinical benefits in comparison with PD-L1 inhibition alone, despite having similar toxicity profiles [66]. Unfortunately, when these preliminary observations were explored in a large, randomized phase III clinical trial, results did not meet the co-primary endpoint of progression-free survival. The study is currently continuing, and further analyses based on overall survival will be further released. Aside from the PD-1 blockade, other immune checkpoints blockades (ICB) combined with TIGIT inhibition also increase anti-tumor immunity. This is mainly because TIGIT has been found to be co-expressed with other immunoregulatory molecules, such as T cell immunoglobulin, mucin domain-containing molecule-3 (TIM-3), and lymphocyte activation gene 3 (LAG-3) [62]. TIGIT^+^ Treg-infiltrating tumors have been found to be characterized by an upregulation of TIM-3, and inhibiting TIM-3 in *Tigit^−/−^* mice further reduces tumor size and increases overall survival [67]. Furthermore, several studies on mice as well as in vitro experimental studies support the notion that dual inhibition of CD112R and TIGIT increases anti-tumor immunity [68]. Curiously, in mouse tumor models, ICB with anti-TIGIT mAbs Fc variants proves that FcγR co-engagement on antigen presenting cells (APCs) increases T cell responses [69]. However, it remains to be clearly demonstrated whether the activity of anti-TIGIT antibodies is Fc-dependent. This question might be answered by the data provided by multiple ongoing phase I and II clinical trials using Fc-engineered anti-TIGIT mAbs. Curiously, in in vitro studies, IL-15 combined with TIGIT inhibition has also been found to increase the activity of NK cells’ cytotoxicity versus that of melanoma cells, and to reduce the development of tumor metastasis in mouse melanoma [70] and soft tissue sarcoma models [71]. These findings might support the development of innovative combination strategies which use IL-15 and TIGIT blockade to induce NK cell-cytotoxicity in patients with abnormalities in MHC-class-I signaling. Taken together, these data suggest that TIGIT might represent a promising target for next-generation cancer immunotherapy. The success of TIGIT inhibition, either alone or in combination, might be largely effective/dependent on NK cell activity. 

In addition to TIGIT, CD155 can also bind to CD96 (tactile), which is a protein from the immunoglobulin superfamily that has been shown, like TIGIT, to compete with CD226 activating receptors for CD155 binding and to promote NK cell inhibition [72]. Similarly to TIGIT, CD96 expression is upregulated during activation and exhaustion on both NK and T cells [60,73] and is frequently co-expressed with other inhibitory receptors, such as TIGIT itself and Lag3 [73,74]. CD96 deficient mice have shown to have reduced carcinogenesis and experimental lung metastasis [72,75]. In a recent study, it was observed that the TIGIT blockade restored NK cell functionality on ovarian cancer patients who displayed altered expression of CD226, TIGIT, and CD96 [76]. In glioblastoma patients, CD96 expression strongly correlated with cancer progression [74]. Although strategies to neutralize the effect of CD96 are not as advanced as those developed for TIGIT, these studies undeniably demonstrate that the DNAM-1/TIGIT/CD96 axis and CD155 immunoregulation represent promising targetable immunotherapeutic strategies for cancer treatment [77,78].

NKG2A/HLA-E: The cell surface molecule NKG2A, a member of a lectin family, is expressed by NK cells and in the tumor microenvironment also by CD8+ cells. It forms a heterodimer with CD94, which is also known as KLRD1, another NK cell-expressed C-type lectin [79]. The engagement of the NKG2A/CD94 complex with its ligand, the non-classical MHC I molecule, (HLA-E in humans and Qa-1b in mice) transduces inhibitory signals, suppressing NK cells’ activity [80]. Recent studies have shown the expression of HLA-E to be upregulated in the TME of several cancer types, including lung, pancreas, stomach, head and neck, and colon cancer [81]. NKG2A is acquired early during NK cell maturation from CD34+ hematopoietic stem cells [82,83]. CD94/NKG2A+ NK cells accounts for more than 50% of circulating NK cells and may express either the CD56bright immature or the CD56dim mature NK cell phenotype [84]. Notably, while CD56bright NKG2A+ cells secrete cytokines, CD56dim NKG2A+ cells also show cytolytic activity and are very effective in removing DCs that fail to achieve adequate maturation [85]. Further studies revealed that, in addition to NK cells, other cell types, including NKT cells [86], CD8+Tαβ cells [87], and Gamma T cells, [88] express the NKG2A/CD94 complex. However, while the NKG2A/CD94 complex is constitutively expressed on most NK cells, its expression on CD8 T cells is upregulated by prolonged antigenic stimulation or by exposure to immunosuppressive cytokines, such as TGF-β [89]. Hence, in contrast to what has been observed for PD-1 and LAG3, which can be considered an early immune checkpoint of CD8+ T cell, the NKG2A receptor is a late-stage immune checkpoint which is expressed upon repeated stimulation and division of CD8+ T cells [90]. Two recently published articles showed that blocking NKG2A can improve the outcome of immunotherapies. The van Hall group demonstrated that the combination of peptide vaccination with antibody inhibition of NKG2A on CD8+ T cells improves response rate and survival of these animals, compared to peptide vaccination alone [91]. Furthermore, the group also demonstrated that it is necessary that tumor cells, but not stromal or immune cells, express Qa-1b (HLA-E homolog in the mouse) for this additive effect. Andre et al. also reported the positive effect of the NKG2A blockade in combination with anti-PD-1/PD-L1 inhibition in murine lymphoma tumor models and in human in vitro experiments [81]. In addition, the preliminary results of a clinical trial in patients with head and neck squamous cell carcinoma showed that the combination of monalizumab, a NKG2A-specific inhibitory humanized antibody, and cetuximab, an anti-EGFR antibody, reached a 30% objective response rate, even in immunotherapy refractory patients [92,93]. The mechanism of action of this combination is most likely mediated by the blockade of the inhibitory NKG2A receptor and the enhancement of cetuximab-based ADCC mediated by the NK cell Fc receptor. While these preliminary results remain to be confirmed, the data from these early trials are notable and emphasize the importance of harnessing anti-tumor activity by means of NKG2A blocking. Interestingly, these results might also suggest that the blockade of NKG2A can not only potentiate the NK and CD8 natural cytotoxicity, but also improve NKs’ ability to kill tumor cells via their potent CD16-mediated ADCC activity when using tumor antigen-specific therapeutic antibodies. Furthermore, a phase I–II trial (NCT02671435) also assessed the effectiveness of monalizumab combined with Durvalumab (anti-PD(L)1) in micro-satellite stable (MSS) CRC patients, thus showing interesting pharmacodynamic immune activation following treatment [94]. The effective activity of the dual immune checkpoint inhibition strategy (anti-PD(L)1 and anti-NKG2A) has also been more strongly confirmed by clinical trials conducted in adjuvant and neo-adjuvant settings in NSCLC patients. Particularly, the anti-PD(L)1 monoclonal antibody (mAb) durvalumab, in combination with the anti-NKG2A mAb monalizumab, has shown interesting response rates and progression-free survival when administered to patients with unresectable, stage III NSCLC in the phase 2 COAST trial (NCT03822351). The mechanism of action leading to these interesting results might possible be associated to an increased activation of CD8+ cells by the dual blockage of PD-L1 and NKG2A [95]. Based on these encouraging results, NeoCoast (NCT05061550), a randomized, open-label, phase 2 study, is currently evaluating the combination of durvalumab with monalizumab as neoadjuvant treatment in patients with resectable, stage IIA–IIIA NSCLC. Following surgery, patients are treated with adjuvant durvalumab plus monalizumab [96]. While these studies remain to be further tested, the preliminary results emphasize the relevance of harnessing the anti-tumor activity of NK cells, when most immuno-therapeutic strategies thus far have focused on the boosting of T cell anti-tumor responses. Surprisingly, NK cells, when unhinged from NKG2A masking, might not only enhance their natural cytotoxicity, but also improve their capacity to destroy tumor cells when tumor antigen-specific therapeutic antibodies are used, via their potent CD16-mediated ADCC activity. 

#### Bispecific and Trispecific Antibodies: Killer Cell Engagers Targeting NK and Tumor Cell Receptors

Recent progress in protein engineering and recombinant DNA technology has enabled the development of bispecific antibodies (BsAbs), consisting of two scFv fragments with different specificities that target either different antigens or different epitopes on the same antigen [97]. Bispecific immunotherapies offer many benefits over monoclonal therapeutics: higher binding avidity and increased cytotoxic effects, as well as improved safety and a lower rate of resistance development due to the simultaneous targeting of two different antigens [98,99]. Among the BsAbs, particular attention is now focused on the BiKEs, which are a type of BsAbs able to redirect autologous immune cells to malignant cells by targeting an immune cell receptor and a tumor antigen, thus creating the immunologic synapse between them and inducing immune cell activation. Most BiKEs have targeted CD16A and trigger potent NK cell activation, thereby inducing their cytotoxic functions [100]. Several CD16A–BiKEs have been developed to target different tumor antigens, including the epithelial cell adhesion molecule (EpCAM) [101], CD33 [102], CD19 [103], or CD20 [104]. Among them is the BiKE AFM13 (Affimed), binding CD16A on NK cells and CD30 on tumor cells, which is currently being evaluated in a phase 1/2 trial in patients with CD30-positive relapsed or refractory Hodgkin and non-Hodgkin lymphomas. The encouraging results of this study have shown a 100% objective response rate and an improvement of complete response rate from 38% to 62% after a second cycle in 13 patients treated with the molecule [105]. In addition to CD16, BiKEs engaging other NK cell receptors such as Nkp30, Nkp46, and NKG2D have been developed [106]. Tri-specific killer cell engagers (TriKEs) targeting NK cells through their activating receptors, Nkp46 and CD16, together with a tumor antigen on the neoplastic cells, also constitutes a new generation of synthetic molecules for cancer immunotherapy. These report superior in vitro efficacy to BiKEs [107]. TriKEs’ functionality can be improved by adding an IL15 moiety. The 161533 TriKE, which incorporates a modified human IL-15 as a crosslinker between the specific scFvs against CD16 and CD33, a myeloid-specific member of the sialic acid-binding receptor family which is highly expressed on myeloid leukemia cells, has demonstrated superior anti-tumor activity both in vitro and in preclinical models compared to the 1633 CD16-CD33 BiKE without IL-15 [108]. This TriKE could also expand and sustain human NK cells in a preclinical model, enhancing its immunotherapeutic potential. Currently, a phase I/II clinical trial of this molecule (NCT03214666) is underway for the treatment of CD33+ myeloid malignancies. A recent study testing the in vitro effectiveness of the Nkp46/CD16A/CD19 TriKE has also shown promising results, supporting its therapeutic use for treatment of refractory/relapsed leukemia [109].

#### Abs Modifying NK Cells’ Activity through an Indirect Effect 

In addition to Abs that directly target NK cell receptors, NK cell activity can be modulated through indirect immune-based mechanisms by some Abs which target receptors in tumor cells associated with anti-apoptotic and proliferative signaling pathways. This is the case for trastuzumab, cetuximab, and rituximab mAbs amongst others.

Trastuzumab: Trastuzumab is a humanized IgG1 mAb that targets the human epidermal growth factor receptor 2 (HER2). It is FDA approved for the treatment of HER2-positive breast and gastric cancers. Aside from inhibiting HER-2 signaling pathways in the malignant cells, anti-HER2 therapies also trigger immune responses that induce indirect tumor cell killing through the induction of NK cell-mediated ADCC [110]. After the IgG1 mAb antigen-binding fragment binds to the extracellular domain of the HER2 surface of the cancer cell, the NK cell surface of CD16A interacts with the Fc portion of the anti-HER2 molecule coating the cancer cell, inducing NK cell activation. This results in both cytotoxicity and cytokine response, enhancing the anti-cancer effects of this molecule [110,111].

Cetuximab: Cetuximab is a human–mouse chimeric IgG1 mAb against the epidermal growth factor receptor (EGFR), which is FDA approved for the treatment of colorectal and head and neck cancer. The efficacy of this molecule is also mediated by its ability to activate NK cells through binding to the therapeutic agent loaded onto EGFR, thus triggering ADCC [112]. Furthermore, cetuximab enhances IFN-γ secretion by activated NK cells, promoting DC maturation and boosting tumor antigen-specific T cell immunity [113].

Rituximab: Rituximab is a human–mouse chimeric IgG1 mAb targeted against the CD20 molecule, which is FDA approved for the treatment of non-Hodgkin’s lymphoma and chronic lymphocytic leukemia. In vitro studies have shown that one of the major mechanisms of action of rituximab is ADCC, highlighting the important contribution of NK cells to the depletion of malignant B cells [114]. This has also been demonstrated in in vivo studies, in which mice deficient in activating Fc receptors did not respond properly to the therapeutic agent [115].

#### 2.3.2. Unleashing NK Cell Activity by Cytokine Therapy

NK cell homeostasis, survival, activation, migration, recruitment, and function are highly influenced by the cytokine environment (immunosuppressive or activating) that surrounds the NK cells. Stimulation of NK cell effector function using cytokines has been one of the earliest and most common approaches to immunoenhancement in cancer therapy. Multiple cytokines have been described in the literature to regulate NK cell immunobiology. These include IL-2 and IL-15, which are implicated in the survival, proliferation, IFN-γ secretion, and cytotoxicity of NK cells [116] as well as IL-12, IL-18, and Type I IFNs (i.e., IFN α/β), which enhance NK cell effector functions [117,118,119]. These cytokines have been widely used to ex vivo expand and produce functional NK cells for NK cell adoptive transfer therapy. On the other hand, it has been broadly described that NK cells, in addition to their cytotoxic capacity, also secrete pro- or anti-inflammatory cytokines, which in turn modulate the adaptive immune response [120]. Therefore, an alternative strategy for cancer immunotherapy would be to focus on the cytokines secreted by NK cells: blocking pro-inflammatory cytokines and/or using methods that increase the secretion of anti-inflammatory cytokines. Similarly, preventing or minimizing NK cell immunosuppression by modulating the presence of suppressor cells within the TME has also been highly explored as a therapeutic strategy to obtain long-lasting, functional NK cells. Tumor cells are known to create a niche that favors the recruitment of regulatory T cells, myeloid-derived dendritic cells, and macrophages with suppressor characteristics that can dampen any anti-tumor effort attempted by immune effector cells, NK cells among them. Consequently, another approach to “push” the activity on NK cells is blocking immunosuppressive cytokines that negatively affect their effector function. Some of the most relevant immunotherapeutic strategies involving cytokines are described below.

#### Blockage of NKs Immunosuppressive Cytokines

TGF-β. TGF-β, usually detected at high levels in the TME, is a suppressive cytokine that inhibits the mTOR signaling pathway in NK cells, leading to their exhaustion [121]. Three TGF-β isoforms have been identified, TGF-β1 being the most prevalent in solid cancers [122]. The inhibitory effects of this cytokine are well-documented and include, among others, the downregulation of the activating receptors NKp30, NKG2D, and CD69 [123] as well as the degranulation marker CD107a, and alterations in the production of the effector cytokines IFN-γ and TNF-α [124,125]. On the other hand, in NK cells, TGF-β signaling also inhibits the expression of T-bet, known to regulate the transcription of genes involved in NK cell cytotoxicity, including Prf1, Gzmb, and Runx1, thus inhibiting Th1 development [126]. Therefore, TGF-β has become an attractive druggable target. Several highly selective anti-TGF-β isoform mAbs have been developed and are under clinical evaluation. Treatment with the murine mAb 1D11 (Genzyme Corp., Sanofi) has shown to neutralize all three isoforms of TGF-β, thus suppressing lung metastases in a murine breast tumor model by increasing infiltration and anti-tumor activity of NK cells and T cells at the metastatic site [127]. In addition, in experimental studies, the combination of 1D11 mAb with IL-2 has allowed the administration of lower non-toxic doses of IL-2, while still maintaining strong anti-tumor responses mediated by both NK and CD8 T cells on tumor mouse models [128]. Furthermore, and given the role of TGF-β on NK cell ontogeny [129], the combination of 1D11 mAb and IL-2 was also able to accelerate NK cell reconstitution in HSCT settings [130]. The humanized version of 1D11, Fresolimumab (GC1008, Genzyme Corp., Sanofi), was tested in a phase I trial on patients with advanced malignant melanoma and renal cell carcinoma (RCC), demonstrating preliminary evidence of anti-tumor activity [131]. Fresolimumab is no longer under consideration for the treatment of cancer, but there is interest in its use against fibrotic diseases [132]. Interestingly, a different version of fresolimumab, which contains a single mutation in the Fc region (SAR439459), has been shown not only to suppress the negative effects of TGF-β on NK and T cells, but also to enhance the anti-tumor response of anti-PD-1 therapy [122,133]. SRK181-mIgG1, a highly selective TGF-β1 mAb inhibitor, could also overcome the resistance to checkpoint inhibitor therapies [134], similarly to SAR439459. Other approaches that have been used to neutralize TGF-β involve the use of small molecules inhibitors that act on the TGF-β receptor kinase activity or TGF-β ligand traps [122]. Recently, a first-in-class bifunctional fusion protein known as bintrafusp alfa (also named GSK-4045154, M7824, and MSB0011359C) has been developed, which consists of an anti-PD-L1 mAb, avelumab, fused with the extracellular domain of the TGF-β receptor to trap soluble TGF-β. This fusion protein has demonstrated clinical activity in several types of cancer in multiple early phase clinical trials (NCT02517398 [135], NCT02699515 [136], NCT02517398 [137]). Interestingly, the treatment of B16F10 in tumor-bearing mice with a bifunctional TGF-β trap/IL-15 protein complex, HCW9218 also enhanced NK and CD8 T cell-mediated anti-tumor responses [138]. These studies support the rationale for immunotherapeutic strategies that combine immune stimulation and neutralization of immunosuppression approaches.

IL-10. NK cells can also be negatively regulated by IL-10 [139,140,141,142,143]. Multiple reports have shown that IL-10 production, mainly by Tregs and MDSCs, can alter NK cell activity and function with downregulation of NKG2D expression, IFNγ production, and degranulation properties, which can be restored by the IL-10 blockade [141,142]. It has also been reported that the lack of NK cell cytotoxicity versus cancer cell with high expression of Hypoxia-inducible gene 2, was shown to be mediated by the IL-10 and STAT3 signaling pathway [144]. Interestingly, a new preclinical study showed that treatment with lipid-protamine-DNA nanoparticles loaded with genes against IL-10 and CXCL12 significantly reduced tumor growth by changing the TME landscape with lower immunosuppressive cells (M2+ macrophages, MDSCs), and activated tolerogenic DC and NK cells [145]. However, given its pleotropic function, IL-10 has also been associated with facilitation of homeostasis and proliferation of immune cells [139]. Indeed, it was shown that human NK cell effector functions can be increased by IL-10 through the regulation of the mTORC1 signaling pathway, which increases the glycolysis and oxidative phosphorylation of NK cells upon IL-10 treatment [146]. Currently, there is a first-in-class, long-acting IL-10 receptor agonist, pegilodecakin, that has entered clinical study when administered as monotherapy or in combination with ICB therapy [139,147]. Unfortunately, little is known about the impact of pegilodecakin on NK cells [148,149,150], thus suggesting a need to conduct additional studies in order to better clarify the immunosuppressive role of IL-10 on NK cells’ activity. 

#### Treatment with Cytokines and Chemokines Inducing NKs Expansion or Recruitment

IL2. IL-2 is known to activate NK cells through recognition of the low affinity dimer formed by IL-2Rβ and the common γ chain [151,152]. Therefore, IL-2 is often the cytokine used for the ex vivo expansion and sustainability of NK cells in NK cell adoptive transfer therapies [153,154,155]. The usage of IL-2 treatment was approved by the FDA in 1992 for RCC and in 1997 for melanoma. IL-2 administration was shown to promote maturation, proliferation, activation, and expansion of NK cells in both preclinical and clinical studies. Unfortunately, in clinical studies, the dosage of IL-2 administered to induce a significant tumor rejection led to severe toxicities, including vascular leak syndrome, heart failure, and liver toxicity in clinical trials. This obstacle, together with the low rates of anti-tumor response, has limited the immunotherapeutic administration of IL-2 as monotherapy [156,157]. In addition, prolonged NK cell stimulation with IL-2 can lead to an exhausted phenotype [60]. Therefore, many efforts have been made to develop alternative therapeutic strategies that can exploit the IL-2 properties while keeping the treatment-related adverse effects at bay. These strategies include, but are not limited to, low dose regimens, synthetic modified IL-2 formulations, and combination with other therapies that modulate molecules known to impact (positively or negatively) NK cell function [158,159]. When given at low doses, IL-2 favors the expansion of regulatory T cells through binding to the high affinity IL-2Rα (CD25), which is constitutively expressed by this immunosuppressive population. Consequently, the IL-2 anti-tumor effect is usually compromised by the suppressive effect that regulatory cells exert on NK and CD8 T cells. Many studies have attempted to improve low dose IL-2 therapeutic effect by depleting regulatory T cells, or neutralizing TGF-β or IL-10 with significant, but minor, increases in the anti-tumor response [128,130,160,161]. Another approach has involved the creation of antibody cytokine immune complexes, which supports the preferential binding of IL-2 to CD122 [162,163]. This was first accomplished by the fusion of IL-2 with the IL-2 mAb clone S4B6, which blocked the recognition of CD25 [162]. Its administration resulted in marked expansion and activation of both NK and CD8 T cells, as well as improvement of the IL-2 half-life [162]. In order to push for its clinical development, a second generation has recently been constructed by splitting and permanently grafting unmutated human IL-2 to its antigen-binding groove on the anti-human IL-2 mAb clone NARA1, which has also demonstrated binding bias towards CD122 [163]. Following the same idea, a bispecific IL-2v immunocytokine, which can only bind to CD122, fused with a fibrobroblast activation protein-α (FAP) named RO6874281. It has entered clinical trials, given its potent anti-tumor response which has been observed in preclinical studies and is currently under evaluation (NCT03875079, NCT02627274) [164,165]. THOR-707, an engineered, semi-synthetic, organism-derived IL-2 variant with lower binding properties to CD25, is also being studied in clinical trials (NCT04009681, NCT05179603).

IL15. The importance of IL-15 for NK cell maturation and survival was demonstrated by the observation that IL-15 deficient mice lacked most of the NK cell population, and wild-type NK cells did not survive in these mice after being adoptively transferred [166]. IL-15, like IL-2, also binds to IL-2Rβ and the common γ chain, but, distinctively, requires its trans-presentation by dendritic cells and stromal cells through the IL-15Rα. Unlike IL-2, it does not directly trigger regulatory T cells [167]. Subcutaneous injections of recombinant human IL-15 (rhIL-15) substantially increased the numbers of circulating NK and CD8 T cells without toxicity in patients with advanced solid tumors [168]. Similarly, the administration of the IL-15/IL-15Rα immune complex induces activation and expansion of both NK and CD8 T cells, which become highly functional with enhanced anti-tumor responses [60,169]. IL-15 has also frequently been used for ex vivo expansion of NK cells. A phase I/II clinical trial probed the tolerability of infused allogeneic IL-15-stimulated NK cells after haplo-Hematopoietic stem-cell transplantation (HSCT) in pediatric patients with refractory solid tumors [170]. One of the most advanced therapies based on IL-15 is N-803 (formerly known as ALT-803), an IL-15 super agonist that consists of a fusion protein encompassing IL-15 and the Sushi domain of IL-15Rα. This compound has demonstrated anti-tumor efficacy in preclinical studies and multiple clinical trials for AML and solid cancer [171,172,173]. N-803 has been frequently combined with haploidentical NK cell adoptive transfer therapy in the allogeneic HSCT settings to support the survival and activation of the infused cells. Its development, efficacy, and evolution have been reviewed elsewhere [174,175]. Surprisingly, it has just recently been reported in two independent clinical trials (NCT03050216 and NCT01898793) that systemic administration of N-803 promoted host CD8 T cell activation, accelerating donor NK cell rejection and, thus, limiting the efficacy of haploidentical NK cell therapy in relapsed/refractory AML patients when compared to IL-2 administration [176]. These results might prompt the redefinition of where and how N-803 can be used in the near future. Other molecules that are being studied include NKTR-255, a novel polyethylene glycol-conjugate of rhIL-15 [177], and NIZ985, a recombinant heterodimer of IL-15/IL15Rα [178], both of which have shown promising anti-tumor efficacy in a preclinical and phase I clinical trial, respectively. 

The administration of plasmid DNA encoding the IL-15 gene has also been shown to promote NK cell expansion and maturation [179]. In order to concentrate IL-15 efficacy within the TME and likely improve its efficacy, intratumoral delivery of plasmid IL-15 DNA has been evaluated in experimental tumor models [180,181]. In addition, in order to facilitate the delivery of the plasmid to the TME, the fusion of apolipoprotein A-I (Apo A-I) with IL-15 has been explored [182]. Ochoa and collaborators created a triple fusion protein that combined Apo A-I with L-15 and IL-15Rα’s sushi domain (Sushi-IL15-Apo), which, at tolerable doses, remarkably enhanced the numbers of CD8 and NK cells, as well as protecting from B16-OVA lung metastasis after hydrodynamic injections [183,184,185]. Moreover, treatment with this protein restored the NK and CD8 T cell compartment in IL-15Rα deficient mice [183]. Another benefit of using the sushi-IL15-Apo protein is the increase in NK cell-mediated ADCC when given in combination with cetuximab against EGFR^+^ tumors [186]. Alternatively, oncolytic virotherapy has also exploited the benefits of IL-15 immunostimulation, with several studies focused on genetically engineered viruses armed with IL-15, either alone or in combination with other molecules, to augment anti-tumor responses [187,188,189,190,191,192]. A similar approach using red blood cells is being tested with the name of RTX-240, a genetically engineered red blood cell expressing 4-1BBL and IL-15/IL-15Rα fusion [193].

IL12. IL-12 is a pro-inflammatory type I cytokine which sustains NK cell survival. Unfortunately, the clinical translation of IL-12-based immunotherapies was hampered by low clinical response and the development of severe clinical toxicities, including deaths of two patients associated with systemic IL-12 injections in early clinical trials [194,195,196]. For this reason, current IL-12 based therapeutic strategies are based on locally targeting IL-12 to the TME, improving its therapeutic efficacy and limiting the toxicities associated with systemic exposure [197,198]. Some of these approaches aim to direct IL-12 localization to the tumor site by promoting in situ overexpression of IL-12. This is performed using plasmid DNA encoding an IL-12 cassette, which has led to anti-tumor responses in preclinical and clinical studies for melanoma and ovarian cancer [197,199,200,201]. A dose escalation phase I clinical trial in patients with recurrent or persistent gynecologic malignances reported partial response and stable disease in 28.6% and 57.1% of the cases, respectively, when patients were treated at the maximum dose studied with an IL-12 plasmid formulated with PEG-PEI-cholesterol lipopolymer (GEN-1) and pegylated liposomal doxorubicin [202]. In vivo electroporation of IL-12 plasmid DNA intra-tumorally has also shown promising results [197]. Similarly, mRNA-based delivery strategies have been utilized to promote IL-12 expression within the tumor sites [203]. Recently, Moderna Therapeutics has reported the preclinical results of a successful novel intratumoral IL-12 mRNA therapy, demonstrating that IL-12 mRNA delivered intratumorally promoted Th1 responses in the TME, thus enhancing anti-tumor immunity [204]. Notably, in this study, the combination of IL-12 mRNA with anti-PD-L1 Ab caused a synergistic anti-tumor effect in tumors that have traditionally responded poorly to PD-1/PD(L)-1 neutralization therapy [204]. Following this success, the intratumoral administration of the lipid nanoparticle (LNP)-formulated mRNA encoding human IL-12, MEDI119, is being evaluated in a dose escalation phase I clinical trial (NCT03946800), either alone or in combination with systemic durvalumab in patients with advanced solid tumors. Preliminary results report tolerability and encourage efficacy [205].

Oncolytic viruses have also been modified to encode IL-12, alone or along with other molecules (GM-CSF or IL-15), in order to promote and enhance both CD8 T and NK cell-mediated anti-tumor responses [147,192,206,207,208,209]. Enhanced response has been shown with IL-12 encoding the Semliki Forest viral vector when combined with the PD-1/PD(L)1 blockade [208].

Positive results using IL-12-Fc, a variant of IL-12, have also been reported in mouse tumor models. The stability of IL-12 in vivo increases with the fusion to an Fc fragment, allowing less frequent dosing and promoting improved NK cell and T cell responses, as well as tumor rejection. Another encouraging strategy is the generation of an IL-12 fusion protein that binds to collagen. By keeping IL-12 within solid tumors, retained by the collagen, the systemic toxicity is minimized. 

Alternatively, in order to improve tolerance and safety of IL-12 therapy, a new strategy based on a necrosis-targeted IL-12 immunocytokine, NHS-IL12, has been developed [210]. NHS-IL12 has been engineered by genetically fusing two human IL-12 heterodimers to the C-termini of the heavy chains of a human IgG1 antibody against DNA/Histone NHS76, which targets exposed dsDNA, for example, in necrotic areas of tumors. In murine xenograft models, NHS-IL12 led to better control of tumor growth [211], and efficacy was improved by local tumor irradiation due to the increase in tumor necrosis [212,213]. A phase I clinical trial has proven the safety of NHS-IL12 as well as an increase in immune cell activity (NK, NKT, and T cells) with a high density of tumor-infiltrating lymphocytes [214]. A recent study demonstrated enhanced anti-tumor efficacy when NHS-IL12 was combined with bintrafusp alfa in a NK and CD8 T cell dependent manner [215].

Chemokines. A major challenge for successful tumor therapy is guiding the immune cells to infiltrate the tumor. Chemokines including CCL2, CXCL9/10/11, and CX3CL1 provide signals for NK cell migration into tissue. Additionally, activated NK cells upregulate expression of the chemokine receptor CXCR3, which controls the NK cell trafficking and accumulation in the tumor tissue through the IFN-γ induced ligands CXCL9, CXCL10, and CXCL11 [216]. However, the challenge remains to increase the expression of these cytokines within the tumor to attract NK cells. Migration of immune cells to the tumor site is controlled by chemokine concentration gradients. A dysregulated expression of chemokines in the TME promotes an array of evasion mechanisms to deflect actual entry of functional anti-tumor effectors, including the local recruitment and proliferation of immunosuppressive cells (e.g., MDSC, Tregs), displacement of effector immune cells, and reductions in their function. These changes in the chemokine levels also represent an important challenge for adoptive cell therapies, limiting the traffic and infiltrations of infused cells into the tumor site. Thus, development of anti-tumor strategies which target chemokine signaling aims to augment NK cell accumulation in the tumor, creating the possibility of improvement of current immunotherapies efficacy [217].

In hematopoietic malignances and solid tumors, an important role for CXCL12 (also known as stromal cell-derived factor 1, SDF-1) and its ligands, CXCR4 and CXCR7, has been shown in the regulation of cell proliferation, survival, chemotaxis, migration, angiogenesis, and adhesion, as well as homing and mobilization of hematopoietic stem cells to the bone marrow [218,219,220]. High levels of these chemokines are correlated with poor prognosis [218,221]. CXCL12 can be expressed by tumor-associated fibroblast, macrophages, or endothelial cells. High levels of CXCL12 are commonly found on typical metastatic sites such as the brain, bone marrow, lymph nodes, and liver. Consequently, it is not strange that CXCR4 and CXCR7 are also found on tumor cells in hematopoietic malignancies and solid tumors such as lung, prostate, and brain cancer [218,221,222]. Thus, targeting the CXCL12/CXCR4/CXCR7 has recently been explored for the treatment of cancer with the development of multiple CXCR12 (30D8) and CXCR4 (MD3100, MSX-122, BPRCX807, WZ811, motixafortide, TN14003, AMD3465, and AMD1170, MDX1338/ulocuplumab) antagonist alternatives that are currently being explored in preclinical and clinical studies [219]. Alternatively, the high expression of CXCL12 in the tumor site can also be used to our advantage. The CXCL12 ligands can be genetically engineered onto effector immune cells in order to foster their migration and homing towards the TME. This is particularly important for NK cell based therapies, because NK cell ex vivo manipulation can negatively alter the expression levels of CXCR4 [223,224]. Following this idea, an approach that forced CXCR4 expression on NK cells was recently used by Levy and collaborators [224]. In this study, the transfection of NK cells with CXCR4^R334X^-coding mRNA prior to NK cell adoptive transfer therapy increased the migratory capacities and homing of NK cells to the bone marrow without altering their phenotype and function [224]. Similarly, Ng and collaborators generated CAR-NK cells that co-expressed anti-B-cell maturation antigen (BCMA) and CXCR4^R334X^ [225]. The adoptive transfer of these genetically modified CAR-NK cells resulted in stronger anti-tumor responses in a xenograft multiple myeloma mouse model, compared to the infusion of the anti-BCMA CAR NK cell alone [225].

#### 2.3.3. NK Cell-Based Therapy: NK Adoptive Transfer and CAR-NK

NK cells have been explored for adoptive immunotherapy in a number of different cancers, particularly in leukemia and lymphoma, given their recognized graft-versus-leukemia (GvL) effect [226,227]. The adoptive transfer of NK cells after ex vivo activation has been shown to be safe and well-tolerated in lymphomas, breast and lung cancer, and metastatic RCC patients [228,229,230]. The adoptive transfer of non-modified or genetically engineered NK cells could provide a more innovative therapeutic approach for cancer immunotherapy.

#### NK Adoptive Transfer Therapy

The adoptive transfer of NK cells entails the transfer to a recipient of NK cell-containing preparations for therapeutic purposes. The NK cells used for infusions might be from a host (autologous) or donor (allogeneic) origin. In the autologous settings, NK cells match patients’ HLA class I and, consequently, are considered a safer choice. However, due to this compatibility, they exert fewer anti-tumor effects against autologous tumors, having reduced cytotoxic capabilities due to lower CD107a degranulation, TNF-a, and IFN-g production [231]. Although new protocols have improved the ability to expand NK cells for adoptive transfer in vivo [232], this approach has had poor efficacy in acute leukemias and in metastatic and non-metastatic solid tumors. This is mainly due to the fact that re-infused autologous NK cells remain in circulation and have limited tumor-homing ability [233]. Additionally, the use of autologous cells as a source for NK cell expansion might represent other challenges related to exhaustion and/or anergy. it has often been noted that cancer patients have dysfunctional NK cells with altered activating patterns and inhibitory receptors [60,234,235], which might or might not be corrected through the ex vivo expansion protocols. 

To increase the NK cell anti-tumor efficacy and prevent NK cell inhibition by self-HLA-I recognizing inhibitory KIRs, autologous NK cells can be selected by function of their KIR expression repertoire. Alloreactive NK cells represent a subset of NK cells that express KIRs for non-self HLA class I, and because they do not bind to the self-HLA-I on the tumor, these cells are readily eliminated. This phenomenon is described above as “missing-self” or “KIR mismatch” [228]. Several strategies have been explored to neutralize the inhibitory impact of KIR on NK cells and improve therapeutic efficacy against HLA-I-expressing tumors. Preclinical studies have shown that the inhibition of KIR2DL1-3 and KIR2DS1-2 by the second generation humanized agonistic mAb lirilumab (IPH2102) increases NK cell cytotoxicity and ADCC in lymphoma, leukemia, and multiple myeloma. It is currently under clinical development [236,237,238]. A recent phase I clinical trial demonstrated safe, but limited, clinical activity of lirilumab when given as a monotherapy [239,240]. Interestingly, dysfunctional NK cells obtained from glioblastoma patients had restored cytotoxic functions and reversed inhibition when treated with lirilumab and IL-2 [241], which indicates that lirilumab might accomplish better results if combined with other strategies that involve NK cell activation.

Due to the suboptimal efficacy of autologous NK cell infusions [242], allogeneic NK cells have been heavily assessed as a source for NK cell adoptive transfer therapy. Allogeneic NK cells can be obtained from peripheral blood, umbilical cord blood, pluripotent stem cells, and commercially available NK cell lines such as NK-92. They can provide an “off-the-shelf” option that can reduce manufacturing costs and dose escalation protocols, and make reproducibility much easier [147,243,244,245]. Early on, Miller and collaborators evaluated the safety and efficacy of haploidentical infusions of NK cells in AML patients with poor prognoses, achieving complete hematologic remission in five out of nineteen patients, partly due to in vivo expansion of donor NK cells [155]. The ensuing studies, which confirmed the consistent benefit of KIR-mismatched NK cells in T cell-depleted haplo-HSCT [246,247], represented a true revolution and paved the way for a new wave of effort in improving the concept of NK cell adoptive transfer. The clinical benefit of using allogenic NK cells can be explained by the enhanced NK cell alloreactivity due to KIR-mismatch. In this context, NK cell alloreactivity may be harnessed in mismatched HLA settings due to KIR ligands, particularly in T cell-depleted haploidentical transplant where T cells have no consistent anti-tumor effect [83,248,249]. Phenotypic identification of the alloreactive NK cell subset and evaluation of their cytolytic activity against leukemic cells represent relevant criteria in suitable donor selection. Multiparametric flow cytometric analysis, using appropriate combinations of mAbs, allows definition and characterization of the size of the alloreactive NK cell population in the donor [250]. Genotypic analysis can be combined with flow cytometry in order to explore suitable donor recipient mismatches and to select appropriate haploidentical HSCT donors [251]. A critical aspect for success is the ability to expand NK cells in culture with irradiated GMP-grade feeder cells expressing IL-15 and 4-1BBL.

Notably, unlike T cell adoptive transfer therapy, the use of allogeneic alloreactive NK cells provides an important advantage, because many studies have suggested the protective role of NK cells against the induction of graft-versus-host disease (GvHD) in allogeneic HSCT [226,230,252,253]. Some of the different mechanisms involved in this protection include the known ability of the NK cells to exert their effector function in a non-antigen specific manner and the regulation of immune cells typically implicated in the GvHD process by different means, all of which have been largely studied and reviewed elsewhere [229,254]. 

Despite the knowledge of the shorter life of adoptively transferred NK cells, which in turn can be an advantage which limits toxicity and off-target effects, in the HLA-haploidentical T cell-depleted NK cell containing HCT, the survival of alloreactive NK cells has been successfully reported to persist for years. This suggests a continuous selection and expansion in vivo [248,255]. NK cells are the first cell type to recover after HCT, and, therefore, may modulate GvHD in addition to their role in the GvL effect. Thus, in T cell depleted haploidentical HSCT, an early GvH and GvL function of adoptively transferred NK cells could provide a bridging function to early NK cell progenies after engraftment with a continuum of patrolling function.

#### CAR-NK Cell Therapy

Genetically engineered NK cells, able to express a chimeric antigen receptor (CAR) molecule and recognize a specific antigen on the tumor cell surface, also represent a powerful cell-based strategy for cancer immunotherapy [256]. Compared to the high number of CAR T therapies, only a few CAR NK trials have been completed thus far. Currently, 23 trials assessing CAR-NK cell therapies for the treatment of solid tumors and hematologic malignancies are registered on ClinicalTrials.gov.

The use of CAR targeting, cell surface-specific, tumor-associated antigens provides a novel methodology to enhance the efficacy of any effector cell. CAR is a genetically engineered protein, composed of an extracellular domain specific for the target antigen (tumoral in this case), which is derived from a single-chain variable fragment (scFv), a transmembrane domain, and, finally, an intracellular signaling domain responsible for the transduction of the activating signal. CAR technology was first applied to T cells, since in vitro expansion and manipulation of these cells were readily available. For this reason, the first clinical trials have already been completed using CAR-T cell therapy [257,258]. In contrast with CAR-T cell technology, the use of CAR-NK constructs has some notable advantages. For instance, CAR-T adoptive transfer is limited to autologous T cells to avoid GvHD, while CAR-NK cells can be generated from autologous and allogeneic sources, given its safety profile regarding GvHD. Additional hurdles for CAR-T cell therapy are represented by manufacturing cost, uncertainty of eventually attainable cell numbers (vs. cell numbers needed), and the time needed for generation and expansion relative to the time-to-relapse or time-to-progression in a patient. On the contrary, CAR-NK cell construction and infusion allow an “off-the-shelf” strategy using previously prepared CAR-NK according to KIR and inhibitory receptor haplotype. As previously stated, in order to fully exploit NK cell capacities, functional NK cells are ideally KIR:HLA class I mismatched to avoid inhibitor receptor blocking. When carrying a CAR construct, this adds major functional advantages by providing activating receptors and lower inhibition thresholds, which can be effective even when the tumor has lost the targeting antigen. Moreover, unlike CAR-T cells, the genetic modification of NK cells with CAR does not seem to alter the repertoire of the activating receptors, and these cells can still recognize and kill tumor cells via the activating NK receptors (e.g., NKp46, NKp30, NKG2D). Another advantage of CAR-NK cells relates to the relatively rare onset of clinically severe cytokine release syndromes (CRS) in treated patients [259], as opposed to more serious and frequent CRS obtained following CAR-T transfer [258,260,261]. 

For these reasons, multiple options are under exploration for the use of CAR-NK cells, due to increasing interest.

So far, and with few exceptions, the predominant number of CAR-NK studies involved NK-92, a transformed cell line known to maintain functional characteristics that can be easily manipulated [244,245,262,263,264]. However, CD16 is not present on NK-92 cells [265], and thus, they cannot be used with mAbs that rely on ADCC function. Nevertheless, the use of NK-92-derived CAR-NK cells has been explored in preclinical studies for the antigen recognition of molecules present on the surface of hematological and solid cancers. These include CD19, CD20, CD138, BCMA, EGFR, HER2, and EpCAM, among many others [225,245,262,266,267,268,269,270,271,272,273,274,275,276,277].

The first clinical application of this approach, conducted in 2018 (NCT02944162), reported the safe infusion of doses up to 5 × 10^9^ of CD33-CAR NK-92 cells into three patients with relapse and refractory (R/R) AML [278]. However, no clinical efficacy was obtained, due to the short-term persistence of the engineered NK cells [278]. A possible explanation for the low response of NK-92-derived CAR-NK might relay on the necessity to irradiate the cells prior to infusion in order to prevent tumorigenesis, thus limiting their in vivo expansion [279]. Currently, there are two more clinical trials awaiting results for multiple myeloma (NCT03940833) and glioblastoma (NCT03383978). In addition, a new NK-92-derived cell line (oNK-1), which endogenously expresses CD16 and is conjugated to trastuzumab, has been created [280]. It has been evaluated with the name of ACE1702 in a phase I clinical trial for advanced solid tumors expressing HER2 (NCT04319757).

The use of primary NK cells from peripheral blood, umbilical cord blood (UCB), or bone marrow sources is preferred due to characteristics of transformed cell lines in in vivo infusion. However, for this purpose, future challenges will entail new, optimized protocols for engineering multiple target CARs, particularly NK cell expansion protocols.

Similarly to NK-92, peripheral blood-derived NK cells also present the handicap of losing CD16 expression during the activation and expansion protocols [281,282], and therefore, other sources for CAR-NK cell therapies might be more suitable. In this regard, stem cell-derived engineered NK cells, from human embryonic stem cells (hESC) or induced pluripotent stem cells (iPSC), provide the advantage of expressing normal levels of activating receptors, CD16 among them. More importantly, these cells allow for a homogenous and renewable “off-the-shelf” approach, as iPSC-derived cell lines are readily available, minimizing donor heterogeneity functions [283,284]. Additionally, because of their low tumorigenicity, these cells do not require irradiation and maintain high cytotoxic functions [283,284]. Exploiting these characteristics, Fate therapeutics (San Diego, CA, USA) is developing several iPSC-derived engineered NK cell products in preclinical and clinical studies. FT596 is a CD19-CAR NK cell with NKG2D and CD3ζ for transmembrane and intracellular domains, respectively, which also contains a non-cleavable CD16 and the recombinant IL-15/IL-15 receptor alpha (rIL-15/IL15Rα). FT596 has shown promising results for the treatment of patients with relapsed refractory B cell lymphomas or CLL, with ten out of fourteen patients reaching objective responses [285].

On the other hand, NK cells obtained from umbilical cord blood (UCB) display a more immature phenotype [286], but stimulation with activating cytokines typically used during the expansion protocols, such as IL-2, are known to provide more functional characteristics [286]. Katy Rezvani’s group from the University of Texas MD Anderson Cancer Center administered UCB-derived CD19-CAR NK product to 11 patients with multiple R/R lymphoid malignancies (NCT03056339). The treatment led to response in eight patients (73%), seven of whom achieved complete remission (63%) without the development of major toxic effects [259]. Remarkably, therapeutic responses were observed within 30 days of the treatment infusion, with CAR-NK persistence for at least 12 months [259]. In this study, an IL-15 construct was also included in the engineered product of the CAR-NK cell, which likely contributed to increased survival of the infused NK cells [259]. Interestingly, the presence of IL-15 seems to be a critical component for NK cell-based therapy success [259,285,287]. A recent study demonstrated improved survival and efficacy of EGFR-CAR-NK cells when an oncolytic virus expressing the human IL15/IL15Rα sushi domain fusion protein was administered at the same time [287]. A higher CAR-NK-mediated efficacy has also been obtained when immunosuppression has been targeted. For example, a recent experimental study has shown that genetically engineered NK cells without the TGFBR2 receptor can break the immunosuppressive milieu conferred by the TGF-β signaling pathway in the brain, allowing them to eliminate the tumor-regenerating glioblastoma stem cells [288]. 

#### Memory-Like NK Cell Therapy

The main hurdle that limits the clinical utility of the adoptive non-engineered NK cell therapies is their lack of in vivo long-term persistence in the absence of cytokine support [289], which can be overcome by generation and use of memory-like NK cells in different settings. Combinatorial pre-activation generates “cytokine-induced memory-like (CIML)” NK cells, which have displayed higher IFNγ production, proliferation, and anti-tumor effects, and share these properties with NKG2C^+^ memory-like NK cells [290,291]. The original observation and characterization of memory-like or adaptive NK cells in a mouse model of cytomegalovirus (CMV) [292] highlighted the possibility that some NK cell subsets may be endowed with some form of antigen specificity [293]. Subsequent work with NK cells expressing CD94/NKG2C showed that memory-like NK cells could also be observed in humans, and represent a critical part of anti-CMV cell-mediated defenses in addition to CMV-specific T cells [294]. It has also been confirmed that a memory-like feature could be added to the generally innate response that characterizes NK cells [295,296]. CMV specificity of human memory-like NKG2C+ NK cells relies on the ability of CD94/NKG2C to bind HLA-E molecules and MHC class I sequence-derived monomers [297]. These observations were groundbreaking for the attempts to generate memory-like NK cells in vitro, leading to the observation that memory-like NK cell functions could be reproduced in vitro when peripheral NK cells were cultured with a cytokine mix containing IL-12+IL-15+IL-18 [298]. These CIML-NK cells demonstrated functional effectiveness in eliminating leukemia cells in vitro in a KIR-independent manner; they also generated a robust response in leukemia patients. In this first-in-human clinical trial, treatment with CIML-NK cells led to clinical responses in five out of nine patients, four of whom reached complete remission (NCT01898793) [298]. In another study, expanded allogeneic NK cells, infused before and after haploidentical HSCT in high-risk myeloid leukemia patients, reduced the relapse rate typically observed after these type of HSCT without adverse events [299]. In addition, a positive trial using systemic IL-15 after NK cell infusion has been reported, with a 32–40% remission rate for relapsed AML after haploidentical transplant, albeit with notable GVHD [173].

Recently, a phase I trial (NCT04024761) with six patients (three AML, one MDS, one CML blast crisis, one blastic plasmacytoid dendritic tumor) reported a response in four patients when CIML-NK cells were adoptively transferred after lymphodepleting chemotherapy and IL-2 administration, in a scenario of post-transplant relapse of myeloid disease [261]. Patients tolerated CIML-NK cell infusion with mild adverse events, namely fever, one case of grade 2 CS, and four cases of pancytopenia, requiring CD34+ cell infusion in two of these cases. Although NK cells expanded efficiently, the highest expansion was observed in the adaptive NK cells in CMV+ recipients. However, analysis of transcription factor expression at day 28 did not reveal significant differences between CMV positive or negative patients, leading to the conclusion that mature CD56dim NK adaptive and non-adaptive NK cells had expanded. In addition, CD56dim NK cells accounted for approximately 35% of all lymphocytes in the peripheral blood after 60 days (versus 7%, pre-infusion), and were found by mass cytometry to phenotypically resemble the corresponding NK cells at screening.

The observation of the long persistence of CIML NK cells recorded in this study contributes to challenging the concept of short-lived NK cells after adoptive transfer therapy. Previous observations indicated that adaptive NK cells with a highly differentiated surface signature (CD94/NKG2C+CD57+selfKIR+ NKG2A-), specific transcription factor modifications, and an HLA-E/CMV-specific response [294,300,301] had a surprising persistence of up to 2 years in the HSCT setting with CMV reactivation [302,303]. In this context, reports showing that CMV-driven expansion of NKG2C^+^CD57^+^ NK cells in HSCT patients correlated with reduced leukemia relapse rates, in addition to the advantage of their longevity, raised the possibility of an immunotherapeutic role for adaptive NKG2C^+^ NK cells against leukemia. [304,305].

As expected, CIML-NK cells have not escaped the trend of CAR-NK cell therapy. A new preclinical study evaluated the efficacy of CAR-NK cells, originating from peripheral blood IL-12/IL-18 CIML-NK cells, against a neoepitope derived from the cytosolic oncogenic nucleophosmin-1 (NPM1) mutated protein to improve the therapeutic option for the treatment of HLA-A2+ AML patients with NPM1c mutations [306]. In xenograft models, NPM1-mutation-specific, TCR-like CAR-NK cells significantly improve the in vivo specific anti-leukemia response against an intracellular mutant protein [306].

## 3. Future Prospects

NK cells quickly destroy malignant cells, making them an optimal target for cancer immunotherapy. However, NK cells play a sort of Cinderella role in immuno-oncology, mainly because the field has sought to boost T cell responses against cancer. It is now clear that many immunotherapies thought to only stimulate T cell responses also activate NK cells, and that NK cells can be beneficial in situations where T cells are not effective. Knowing this, more research is necessary to fully utilize the power of the immune system accurately and effectively, including NK cells, against cancer.

## Figures and Tables

**Figure 1 cells-11-03147-f001:**
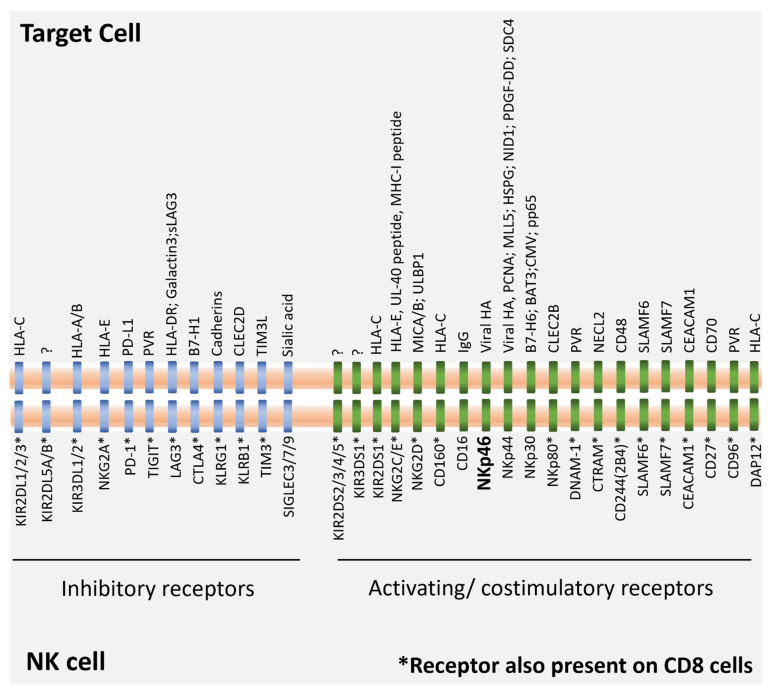
Repertoire of NK cell receptors. Inhibitory receptors are represented in blue, while activating/costimulatory receptors are represented in green. Apart from NKp46, receptors regulating NK cells’ activity are also expressed by additional immune cells, particularly CD8.

**Figure 2 cells-11-03147-f002:**
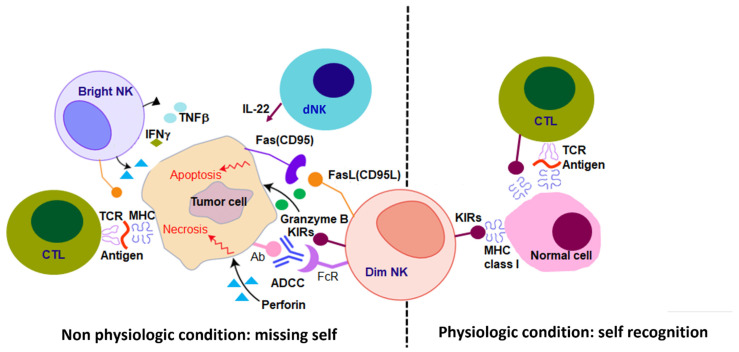
Two main mechanisms lead to NK-mediated cytotoxic activity of target cells: i) the release of cytoplasmic granules containing perforin and granzyme; and ii) the expression of tumor necrosis factor (TNF) family members, such as FASL or TNF-related apoptosis-inducing ligand (TRAIL), which induce tumor cell apoptosis. Moreover, most NK cells express the low-affinity activating receptor FcγRIIIa (CD16), which binds the Fc portion of immunoglobulin G1 (IgG1) and mediates antibody-dependent cellular cytotoxicity (ADCC). In physiologic conditions, a ‘self-recognition’ phenomenon is occurring, during which the binding of NK inhibitory receptor with self MHC class I allows the downregulation of the activating receptors and the tolerance of NK cells to self-tissue. In non-physiologic conditions, such as the insurgence of tumor malignancies or bacterial/viral infections, a ‘missing self’ phenomenon is instead observed, which is characterized by a downregulation or deletion of MHC class I molecules, a blockage of inhibitory signals, and the induction of NK cell activating and cytotoxic functions. Abbreviations: Bright NK, CD56 bright NK cells; CTL, cytotoxic T lymphocyte; Dim NK, CD56 dim NK cells; IFN-γ, interferon-γ; IL-22, interleukin-22; KIRs, Killer Immunoglobulin-like Receptors; MHC, major histocompatibility complex; TCR, T cell receptor; TNF-β, tumor necrosis factor-β; dNK, decidual NK cells.

**Table 1 cells-11-03147-t001:** Summary of selected clinical studies assessing the relationship between circulating and infiltrated NK cells and the clinical outcome in different selected cancer types. MM: multiple myeloma; NSCLC: non-small cell lung cancer; SKCM: skin cutaneous melanoma; HNSC: head and neck squamous cell carcinoma; breast; CRC: colorectal cancer; BR: brain tumors. ++: very positive association; +: positive association; NA: not applicable; BM: bone marrow; FFPE: formalin-fixed paraffin-embedded; IHC: immunohistochemistry; PB: peripheral blood; PBMC: peripheral blood mononuclear cells; TCGA: the cancer genome atlas. ↑increased number of selected cells; ↓ reduced number of selected cells.

NK Localization	Cancer Type							Technical Information				
	MM	NSCLC	SKCM	HNSC	Breast	CRC	BR	Sample Type	NK Cell Markers	Factors Associated with Better Prognosis	Detection Method	References
Periphery	++	NA	NA	NA	NA	NA	NA	PBMC	CD56, NKp30, NKp44, NKp46, NKG2D, CD16, 2B4/CD244	↑ CD16 NK cells ↑ 2B4/CD244 NK cells	Flow cytometry	Fauriat et al., 2006 [34]
	NA	++	NA	NA	NA	NA	NA	PBMC	CD56, CD16, NKp30, NKp44, NKp46, NKG2D	↓ NKp46^+^ CD56^dim^ CD16^+^ NK cells	Flow cytometry	Picard et al., 2019 [20]
	NA	NA	++	NA	NA	NA	NA	PBMC	CD56, CD16, NKp46, NKG2D, NKG2A, CD95, CD11a, CD38, PD-1, CD158b, KLRG1	↓ CD56^bright^ NK cells	Flow cytometry	De Jonde et al., 2019 [18]
	NA	NA	NA	NA	++	NA	NA	PBMC	CD56	↓ total NK cells	Flow cytometry	Larsson et al., 2022 [35]
	NA	NA	NA	NA	NA	++	NA	PB	CD56, CD16	↑ % NK cells	Flow cytometry	Tang et al., 2020 [17]
	NA	NA	NA	NA	NA	NA	++	PBMC	CD56, CD16, NKp30, NKp44, NKp46, NKp80, NKG2D, DNAM-1	↓ *NKp30 i3* expression	Flow cytometry RT-PCR	Semerano M et al., 2015 [27]
TME	++	NA	NA	NA	NA	NA	NA	BM	CD56, CD57, KIR2DL1/S1, CD69, CD16, DNAM-1, NKG2D, SLAMF7, CD11a, NKp30, NKp46, NKp44	↓ SLAMF7 NK cells	Flow cytometry	Pazina T et al., 2021 [36]
	NA	++	NA	NA	NA	NA	NA	FFPE tissues	CD57	↑ NK cell infiltration	IHC	Villegas et al., 2002 [32]
	NA	NA	++	NA	++	NA	NA	In silico analysis (TCGA)	*NCR1*, *KLRF1*, *KLRD1*, *PRF1*, *FCGR3A*, *CCL4*, *CCL3*, *CD247*	↑ expression NK cell specific signature	scRNAseq RNAseq Microarray	Ascierto et al., 2019 [30]
	NA	NA	NA	++	NA	NA	NA	FFPE tissues	CD57	↑ CD57^+^ NK cell infiltration	IHC	Fang et al., 2017 [37]
	NA	NA	NA	NA	++	NA	NA	Frozen tissues, FFPE tissues	*CRTAM*, *CD96*, *CD1d*, *LFA-1*, *CD56*, *CD16*, *NKG2D*, *DNAM1*, *NKp30*, *NKp44*, *NKp46*	↑ *NKp30*, *NKp46*, *NKG2D*, *CRTAM* DNAM1, *CD96* expression	Microarray analysis RT-PCR	Ascierto et al., 2013 [29]
	NA	NA	NA	NA	NA	++	NA	FFPE tissues	CD57	↑ NK cell infiltration	IHC	Coca et al., 1997 [31]
	NA	NA	NA	NA	NA	NA	++	FFPE tissues	Nkp46	↑ NK cell infiltration	IHC	Melaniu et al., 2020 [38]

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
