# Peer review of "Next Generation Immuno-Oncology Strategies: Unleashing NK Cells Activity"

_cells, 2022, doi:10.3390/cells11193147_

Round 1

Reviewer 1 Report

This work is an extensive, lengthy review of NK cells in the context of cancer immunotherapy.  The coverage is excellent, with adequate citations. As the focus is cancer immunotherapy, the title should be changed to indicate that autoimmune immunotherapy is not directly addressed, so that those readers primarily interested in cancer will be drawn to the manuscript.  Line 206 (or maybe line 149 would be a better spot) should start a new major section on cancer therapy---the previous text is a good introduction to NK cell biology, and should be separate. The organization after this point is nice (line 225--antibody treatment, line 479--cytokine therapy, line 741--Car-NK and NK adoptive transfer therapy make for good subtopics).

minor concerns:

line 416--please add "Bispecific antibodies" to the title somehow for 1.b

line 512--missing final word? "....to mediate (what?)"

line 505--"inhibits" not "inhibit"

Reviewer 2 Report

The paper is usefull and interesting.

Table 1 is corrupted in the version or review. I do not understand the color code in Fugure 1.

It would be helpful if the authors could add a paragraph on the role of NK cells in alloSCT including GvH and GvL, and discuss the developments in KIR typing /matching in alloSCT.
